

# Characterization of the mitochondrial genome of *Analcellicampa xanthosoma* gen. et sp. nov. (Hymenoptera: Tenthredinidae)

Gengyun Niu[1,*], Yaoyao Zhang[1,*], Zhenyi Li[2] and Meicai Wei[1]

[1] College of Life Sciences, Jiangxi Normal University, Nanchang, Jiangxi, China
[2] Bangor College, Central South University of Forestry and Technology, Ministry of Education, Changsha, Hunan, China
[*] These authors contributed equally to this work.

## ABSTRACT

A new genus with a new species of the tribe Hoplocampini of Hoplocampinae was described from China: *Analcellicampa xanthosoma* Wei & Niu, **gen. et sp. nov.** *Hoplocampa danfengensis* G. Xiao 1994 was designated as the type species of the new genus. The characters of *Analcellicampa danfengensis* (G. Xiao) **comb. nov.** were briefly discussed. A key to the tribes and known genera of Hoplocampinae was provided. The nearly complete mitochondrial genome of *A. xanthosoma* was characterized as having a length of 15,512 bp and containing 37 genes (22 tRNAs, 13 protein-coding genes (PCGs), and 2 rRNAs). The gene order of this new specimen was the same as that in the inferred insect ancestral mitochondrial genome. All PCGs were initiated by ATN codons and ended with TAA or T stop codons. All tRNAs had a typical cloverleaf secondary structure, except for *trnS1*. Remarkably, the helices H991 of *rrnS* and H47 of *rrnL* were redundant, while helix H563 of *rrnL* was highly conserved. A phylogeny based on previously reported symphytan mitochondrial genomes showed that *A. xanthosoma* is a sister group to *Monocellicampa pruni*, with high support values. We suggest that *A. xanthosoma* and *M. pruni* belong to the tribe Hoplocampini of Hoplocampinae.

# INTRODUCTION

The Nematinae s. lat. (sensu *Prous et al., 2014*) is the second-largest subfamily within the Tenthredinidae and includes approximately 1,260 known species (*Taeger, Blank & Liston, 2010*; *Wei & Niu, 2011*; *Li & Wei, 2012*; *Wei & Xia, 2012*; *Liu, Niu & Wei, 2018*; *Liu, Liu & Wei, 2017*; *Liu et al., 2018a*; *Liu et al., 2018b*).

The systematic arrangement of the genera of Nematinae s. lat. has not yet reached consensus, and approximately ten different systems have been proposed since *Ashmead (1898a)*. Before *Ashmead (1898a)*, the taxon was regarded as a subfamily of Tenthredinidae by *Kirby (1882)*, *Cameron (1882)* and *Dalla Tolle (1894)*. *Ashmead (1898a)* first established a family for Nematinae auct. except Hoplocampinae, which he placed into Selandriidae (*Ashmead, 1898b*). Ashmead's opinion was overlooked by many Symphyta researchers,

Corresponding author
Meicai Wei, weim@jxnu.edu.cn

who treated Nematinae s. lat. as a normal subfamily of Tenthredinidae (*Macgillivray, 1906*; *Benson, 1938*; *Takeuchi, 1952*; *Benson, 1958*; *Malaise, 1963*; *Zhelochovtsev, 1988*; *Taeger, Blank & Liston, 2010*; *Prous et al., 2014*). Seldom has the taxon been treated as a tribe of Tenthredininae (*Enslin, 1918*) or even a subtribe of Tenthredinini (*Konow, 1905*). A few researchers separated Nematinae s. lat. into Nematinae s. str. and Hoplocampinae (*Rohwer, 1911*), Susaninae and Nematinae (*Ross, 1951*; *Maxwell, 1955*; *Smith & Burks, 1979*; *Abe & Smith, 1991*), or Cladiinae and Nematinae (*Ross, 1937*) or regarded it as Nematidae, a family of Tenthredinoidea s. str., with 3 subfamilies: hoplocampinae, Susaninae and Nematinae s. str. (*Wei & Nie, 1998*).

*Nyman et al. (2006)* analyzed the internal phylogeny of Nematinae s. lat. based on the *COI* gene and the nuclear elongation factor-1α gene. The branches (*Craterocercus* + (*Susana*, *Hoplocampa*)) and (*Cladius* + (*Priophorus*, *Trichiocampus*)) were the two most basal paraphyletic clades.

Based on a molecular phylogenetic study, *Prous et al. (2014)* proposed the latest system for Nematinae s. lat. The authors regarded (((*Susana*, *Hoplocampa*) *Cladius*) + *Craterocercus* + *Monocellicampa*) as a middle monophyletic clade of Nematinae s. lat. based on Bayesian analysis or ((*Susana*, *Hoplocampa*) (*Moricella*, *Cladius*)) as a basal monophyletic clade of Nematinae s. lat. based on maximum likelihood (ML) analysis. The branch of (*Susana* + *Hoplocampa* + *Cladius*) of Nematinae s. lat. is most likely a monophyletic group, although the systematic position and the members of the branch are uncertain: the position of *Monocellicampa*, *Craterocercus* and *Moricella* is questionable. Importantly, the cladograms of Nematinae s. lat. in *Prous et al. (2014)* were reconstructed based only on 400-bp sequences of the barcode region.

Based on a comparison of external morphology, *Analcellicampa* Wei & Niu **gen. nov.** and *Monocellicampa* Wei are undoubtedly members of *Hoplocampini*, and they are quite closely related to *Hoplocampa*, as shown by the following characteristics: head strongly flattened; mandibles short and hardly bent, symmetrically tridentate; epicnemium of the mesepisternum flat, with the epicnemial suture fine and vestigial; antenna short and filiform with two basal antennomeres much longer than broad; anal cell in the forewing with a long constricted petiole near the basal 0.4, vein 2r present, 1m-cu meeting cell 1Rs and 2m-cu meeting cell 2Rs; and radix of the lancet and lance quite short. *Analcellicampa* and *Monocellicampa* are probably sister groups within Hoplocampini, as the two genera share a peculiar synapomorphic character with the Nematidae of *Wei & Nie (1998)*: cell M in the hind wing open. *Caulocampus* (*Rohwer, 1912*) and *Armenocampus* (*Zinovjev, 2000*) are two closely related genera and probably also members of Hoplocampini. In addition, Anhoplocampini, including *Anhoplocampa* (*Wei, 1998*) and *Zhuangzhoua* (*Liu, Liu & Wei, 2017*), is regarded as a tribe of Hoplocampinae and related to Cladiini (*Liu, Liu & Wei, 2017*).

A broadened Hoplocampinae subfamily based on *Wei (1998)*, *Wei & Nie (1998)* and *Prous et al. (2014)* is followed tentatively in this paper. The tribes of the subfamily are rearranged. However, there is a strong possibility that Susanini, Hoplocampini, Anhoplocampini and Cladiini are basal paraphyletic groups of Nematinae s. lat. (=Nematidae sensu *Wei & Nie, 1998*).

Here, to confirm the sister group relationship between *Monocellicampa* and *Analcellicampa* and to help clarify the systematic position of *Analcellicampa* within Tenthredinidae, the mitochondrial genome of *Analcellicampa xanthosoma* was sequenced and phylogenetically analyzed, and the new genus and new species were described. For comparison and identification of the members of Hoplocampinae, a key to the tribes and known genera of Hoplocampinae was also provided.

## MATERIALS & METHODS

### Description of the new species

Insect specimens were examined using a Leica S8APO dissecting microscope. Detailed images of adults were taken with a Leica Z16 APO/DFC550 and were then montaged using Helicon Focus (Helicon Soft). Montaged images were further processed using Adobe Photoshop CS 6.0.

The terminology used for sawfly genitalia follows *Ross (1945)*, whereas the terminology for general morphology follows *Viitasaari (2002)*. Abbreviations are as follows: OOL = distance between the eye and outer edge of lateral ocelli; POL = distance between the mesal edges of lateral ocelli; and OCL = distance between the lateral ocellus and the occipital carina or hind margin of the head.

The holotype and all paratypes of the new species were deposited in the Asian Sawfly Collection (ASC), Nanchang, China.

The electronic version of this article in portable document format (PDF) will represent a published work according to the International Commission on Zoological Nomenclature (ICZN), and hence, the new names contained in the electronic version are effectively published under the code from the electronic edition alone. This published work and the nomenclatural acts it contains have been registered in ZooBank, the online registration system for the ICZN. The ZooBank Life Science Identifiers (LSIDs) and associated information can be viewed through any standard web browser by appending the LSID to the prefix http://zoobank.org/. The LSID for this publication is urn: lsid: zoobank. org: pub:8093814F-7A3A-460B-A590-4EF928A280E1. The online version of this work is archived and available from the following digital repositories: PeerJ, PubMed Central and CLOCKSS.

### DNA library construction and sequencing

Total DNA was extracted from *A. xanthosoma* using an E.Z.N.A.® Tissue DNA Kit (Omega, Norcross, GA) following the manufacturer's instructions and stored at −20 °C. Sequencing libraries with approximately 400-bp insertions were constructed using a NEXTflex™ Rapid DNA-Seq Kit (Illumina, San Diego, CA) according to the manufacturer's protocol. Each library was sequenced on an Illumina HiSeq X Ten to generate 150-bp paired-end reads.

### Mitogenome assembly

Next-generation sequencing and bioinformatic analyses were performed by Shanghai Majorbio Bio-pharm Technology Co., Ltd. Reconstruction of the mitochondrial genome from Illumina reads was carried out using three different approaches to ensure the

accuracy of the assemblies: SOAPdenovo v2.0 (*Luo et al., 2012*), MITObim v1.8 (*Hahn, Bachmann & Chevreux, 2013*) and NOVOPlasty v2.7.1 (*Dierckxsens, Mardulyn & Smits, 2017*). The assembled mitochondrial fragments were identified by BlastX using *Asiemphytus rufocephalus* (KR703582) mitochondrial genes as query.

## Mitogenome annotation and secondary structure prediction

All RNA genes were identified by employing the online MITOS tool (http://mitos.bioinf. uni-leipzig.de/index.py) (*Bernt et al., 2013*) with the invertebrate mitochondrial genetic code. The initiation and termination codons of protein-coding genes (PCGs) were determined by Geneious v8.0.5 (http://www.geneious.com) using reference sequences from other symphytan species with subsequent manual adjustment. The A + T content of nucleotide sequences and relative synonymous codon usage (RSCU) were calculated using MEGA v7.0 (*Kumar, Stecher & Tamura, 2016*). Strand asymmetry was calculated using the formulas (*Perna & Kocher, 1995*) GC-skew $= (G - C)/(G + C)$ and AT-skew $= (A - T)/(A + T)$ for the strand encoding the majority of the PCGs.

The secondary structures of *rrnS* and *rrnL* were partitioned into four areas and six areas, respectively. Secondary structures of rRNAs were inferred using alignment to models predicted for *Trichiosoma anthracinum* and *Labriocimbex sinicus* (YC Yan, 2019, unpublished data). First, the primary sequence and the secondary structure of these two species were aligned in MARNA (*Siebert & Backofen, 2005*) to identify a consensus sequence as well as a consensus structure in the output files. Second, the secondary structures of *rrnS* and *rrnL* in *A. xanthosoma* were predicted by specific structural models in SSU-ALIGN (*Nawrocki, 2009*). Finally, the structures were artificially transformed into their relative secondary structures with minor changes.

The predicted secondary structures of RNAs were drawn using VARNA v3-93 (*Darty, Denise & Ponty, 2009*) and RnaViz 2.0.3 (*De Rijk, Wuyts & De Wachter, 2003*). Helix numbering follows that of *Apis mellifera* (*Gillespie et al., 2006*), with minor modifications.

## Phylogenetic analysis

We used Maximum Likelihood (ML) and Bayesian inference (BI) methods to construct phylogenetic trees of selected species using 13 PCGs and two rRNAs (Table 1). The mitochondrial genome sequences of the selected species were downloaded from GenBank. Thirteen PCGs were aligned individually by MUSCLE in MEGA v7.0, and two rRNAs were aligned by MAFFT (https://www.ebi.ac.uk/Tools/msa/mafft/) (*Katoh & Standley, 2013*). Then, the aligned nucleotide sequences were concatenated with SequenceMatrix v1.7.8 (*Vaidya, Lohman & Meier, 2011*) and partitioned into several data blocks.

The partitioned data block file was used to infer both partitioning schemes and substitution models in PartitionFinder v1.1.1 (*Lanfear et al., 2012*) with "unlinked" branch lengths under the "greedy" search algorithm. The standard partitioning schemes "bic" and "aicc" were selected for BI and ML analyses, respectively. BI analyses were conducted with the GTR + I + G model and HKY+G model using MrBayes v3.2.2 (*Ronquist et al., 2012*). Four simultaneous Markov chains (three cold, one heated) were run for two million generations in two independent runs, with sampling every 1,000 generations and 25% of the first generations discarded as burn-in.

**Table 1** Summary information of symphytan mitochondrial genomes used in phylogenetic analyses.

|  | Species | Family | Accesion number | References |
|---|---|---|---|---|
| Ingroup | *Analcellicampa xanthosoma* | Tenthredinidae |  | This study |
|  | *Chinolyda flagellicornis* | Pamphiliidae | MH577057 | *Niu et al. (2019)* |
|  | *Megalodontes cephalotes* | Megalodontesidae | MH577058 | *Niu et al. (2019)* |
|  | *Megalodontes spiraeae* | Megalodontesidae | MH577059 | *Niu et al. (2019)* |
|  | *Allantus luctifer* | Tenthredinidae | KJ713152 | *Wei, Niu & Du (2014)* |
|  | *Arge bella* | Argidae | MF287761 | *Du et al. (2018)* |
|  | *Asiemphytus rufocephalus* | Tenthredinidae | KR703582 | *Song et al. (2016)* |
|  | *Calameuta filiformis* | Cephidae | KT260167 | *Korkmaz et al. (2016)* |
|  | *Calameuta idolon* | Cephidae | KT260168 | *Korkmaz et al. (2016)* |
|  | *Cephus cinctus* | Cephidae | FJ478173 | *Dowton et al. (2009)* |
|  | *Cephus pygmeus* | Cephidae | KM377623 | *Korkmaz et al. (2015)* |
|  | *Cephus sareptanus* | Cephidae | KM377624 | *Korkmaz et al. (2015)* |
|  | *Characopygus scythicus* | Cephidae | KX907848 | *Korkmaz et al. (2018)* |
|  | *Labriocimbex sinicus* | Cimbicidae |  | YC Yan, 2019, unpublished data |
|  | *Corynis lateralis* | Cimbicidae | KY063728 | *Doğan & Korkmaz (2017)* |
|  | *Hartigia linearis* | Cephidae | KX907843 | *Korkmaz et al. (2018)* |
|  | *Janus compressus* | Cephidae | KX907844 | *Korkmaz et al. (2018)* |
|  | *Monocellicampa pruni* | Tenthredinidae | JX566509 | *Wei, Wu & Liu (2015)* |
|  | *Birmella discoidalisa* | Tenthredinidae | MF197548 | GY Niu, 2017, unpublished data |
|  | *Orussus occidentalis* | Orussidae | FJ478174 | *Dowton et al. (2009)* |
|  | *Pachycephus cruentatus* | Cephidae | KX907845 | *Korkmaz et al. (2018)* |
|  | *Pachycephus smyrnensis* | Cephidae | KX907846 | *Korkmaz et al. (2018)* |
|  | *Perga condei* | Pergidae | AY787816 | *Castro & Dowton (2005)* |
|  | *Syrista parreyssi* | Cephidae | KX907847 | *Korkmaz et al. (2018)* |
|  | *Tenthredo tienmushana* | Tenthredinidae | KR703581 | *Song et al. (2015)* |
|  | *Trachelus iudaicus* | Cephidae | KX257357 | *Korkmaz et al. (2017)* |
|  | *Trachelus tabidus* | Cephidae | KX257358 | *Korkmaz et al. (2017)* |
|  | *Trichiosoma anthracinum* | Cimbicidae | KT921411 | *Song et al. (2016)* |
|  | *Taeniogonalos taihorina* | Trigonalidae | NC027830 | *Wu et al. (2014)* |
|  | *Parapolybia crocea* | Vespidae | KY679828 | *Peng, Chen & LI (2017)* |
| Outgroup | *Paroster microsturtensis* | Dytiscidae | MG912997 | *Hyde et al. (2018)* |
|  | *Neopanorpa phlchra* | Panorpidae | FJ169955 | J Hua, 2016, unpublished data |
|  | *Neochauliodes parasparsus* | Corydalidae | KX821680 | *Zhao, Zhang & Zhang (2017)* |
|  | *Anopheles gambiae* | Culicidae | L20934 | *Beard, Hamm & Collins (1993)* |

ML analyses were conducted with the GTR + I + G, GTR + G and HKY + G models. With the best-fit model of nucleotide substitution, phylogenetic construction based on ML was performed on the IQ-TREE web server (http://iqtree.cibiv.univie.ac.at/). The previous data block file as well as the original parameters was also used. In addition, 0.1 was employed as the disturbance intensity, and 1,000, as the IQ-TREE stopping rule.

All related files have been uploaded to figshare (http://figshare.com/s/5d9c3789708b3ebdbe2c).

# RESULTS AND DISCUSSION

## Description

### *Analcellicampa* Wei & Niu, gen. nov.

urn: lsid: zoobank. org: act: FFD0BF94-EF5B-4BE1-8943-72A766426E9A

Type species: *Hoplocampa danfengensis* G. *Xiao, 1994*.

Description. Body small, not slender (Fig. 1). Head strongly compressed in lateral view (Fig. 2C); clypeus broad, flat, anterior margin shallowly incised (Figs. 2B and 3C); eyes large, inner margins convergent downwards, shortest distance between eyes longer than longest axis of eye (Figs. 2B and 3C); malar space as long as or shorter than diameter of median ocellus (Figs. 2B and 3C); mandibles short, evenly tapering toward apex, subsymmetrically tridentate, middle tooth sharp, lowest tooth obtuse (Figs. 4G and 4H); maxillary palp with 6 palpomeres, palpomere 3 not enlarged (Fig. 4C); labial palp with 4 palpomeres, palpomere 3 roundish (Fig. 4D); middle fovea absent, lateral fovea small but distinct (Fig. 2B); hind orbit round, postorbital groove and occipital carina absent; frons weakly elevated, frontal wall indistinct; ocellar triangle very low; postocellar area very short and broad, approximately 3–4.5 times as broad as long; in dorsal view, temple very short and strongly narrowed behind eyes (Figs. 2A and 3F); antenna short and slender, not longer than head and thorax together, much shorter than vein C of forewing, scape and pedicel much longer than broad, antennomere 3 clearly longer than antennomere 4, antennomeres 5–8 each less than 3 times as long as broad (Figs. 2L and 4I); mesepisternum roundly and weakly elevated, epicnemium flat, largely glabrous, epicnemial suture fine but distinct, not furrow-like (Fig. 2F); inner apical spur of fore tibia not bifurcate, with a distinct membranous lobe reaching halfway to apex (Fig. 4E); hind tibia as long as or slightly longer than hind tarsus, inner apical spur of hind tibia shorter than apical breadth of tibia; metabasitarsus as long as following 3 tarsomeres together; tarsal pulvilli distinct (Fig. 4J); claw small, roundly bent, basal lobe and inner tooth absent (Figs. 2J and 4F). Forewing (Fig. 4A): veins 1M and 1m-cu strongly convergent toward pterostigma, vein R +M not shorter than cu-a, first abscissa of vein Rs entire but weak, cell 1Rs shorter than cell 2Rs, vein 2r present, 2r and 2m-cu meeting cell 2Rs, vein cu-a meeting cell 1M near middle, anal cell broadly constricted at approximately basal third with a long middle petiole, basal anal cell closed. Hindwing (Fig. 4B): cells R1 and Rs closed, cells M and A open, vein 2A very short and approximately 1.5–3 times length of vein cu-a; cercus slender, approximately 3–10 times as long as broad (Figs. 2H and 3E); ovipositor sheath shorter or longer than hind femur, apical sheath approximately as long as or longer than basal sheath, tapering toward apex (Figs. 2G and 3E); lancet weakly sclerotized, ctenidium and spiculella absent, serrulae oblique with fine subbasal teeth, radix less than 0.3 times total length of lancet (Figs. 2K and 3G); gonocardo quite narrow at middle and distinctly broadened laterally, middle breadth approximately 2.5 times breadth of thinnest lateral arm (Fig. 3I); penis valve with a small apical lobe, without stout valvispina, surface of valviceps with many small teeth (Fig. 3K); harpe longer than broad (Fig. 3I).

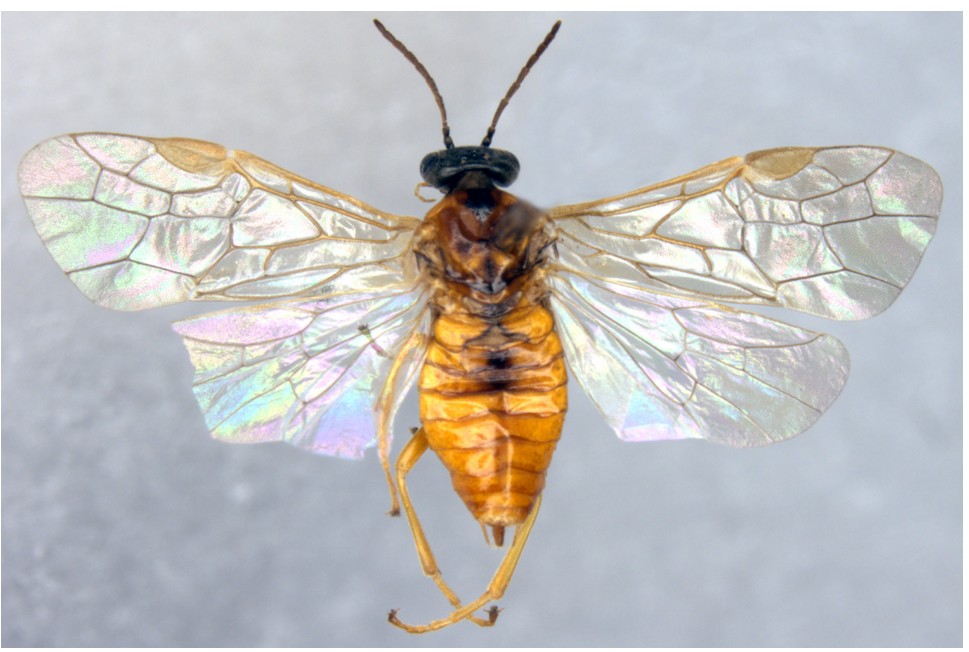

**Figure 1** *Analcellicampa xanthosoma.* Adult female, dorsal view. Scale bar = 1 mm. Photos: Yaoyao Zhang.

Etymology. The generic name, *Analcellicampa*, is composed of *Anal-* and *-cellicampa*, with the former referring to the open hind anal cell of the new genus and the latter referring to a part of the name of *Monocellicampa*, the nearest relative of *Analcellicampa* **gen. nov.**

**Distribution.** China.

**Host plant**: *Cerasus* spp. of Rosaceae. Larvae are borers of the fruits of *Cerasus* spp.

**Remarks**. This new genus is a member of Hoplocampinae (*Wei, 1998*; *Wei & Nie, 1998*) and allied to *Monocellicampa Wei, 1998* but differs from it by the following: the anal cell of hind wing broadly opened at apex, vein 2A very short; epicnemium large with a distinct epicnemial suture; head strongly compressed in lateral view; the inner tibial spur of foreleg simple with a membranous lobe far from apex; the gonocardo quite narrow; the valviceps of penis valve without a stout valvispina but with many small surface teeth; larvae feed on *Cerasus* spp. In *Monocellicampa*, the anal cell of hind wing closed, vein 2A as long as 1A; epicnemium vestigial with epicnemial suture indistinct; head weakly compressed in lateral view; the inner tibial spur of foreleg bifurcate at apex; the gonocardo broad; the valviceps of penis valve with a stout valvispina and without small surface teeth; larvae feed on *Prunus* spp.

*Analcellicampa xanthosoma* **Wei & Niu, sp. nov.** (Figs. 1–2)

urn: lsid: zoobank. org: act: 4EB9F310-6479-4363-A481-81ECF9EF18B6

Female. Body length 5 mm (Fig. 1). Body yellow-brown; head black, clypeus and mandibles largely black, labrum and palps pale brown (Fig. 2B), antennal flagellum dark brown to black-brown (Fig. 2L); anterior 0.3 of propleuron black-brown; center of mesoscutal middle lobe, top of mesoscutal lateral lobe, posterior of mesoscutellum,

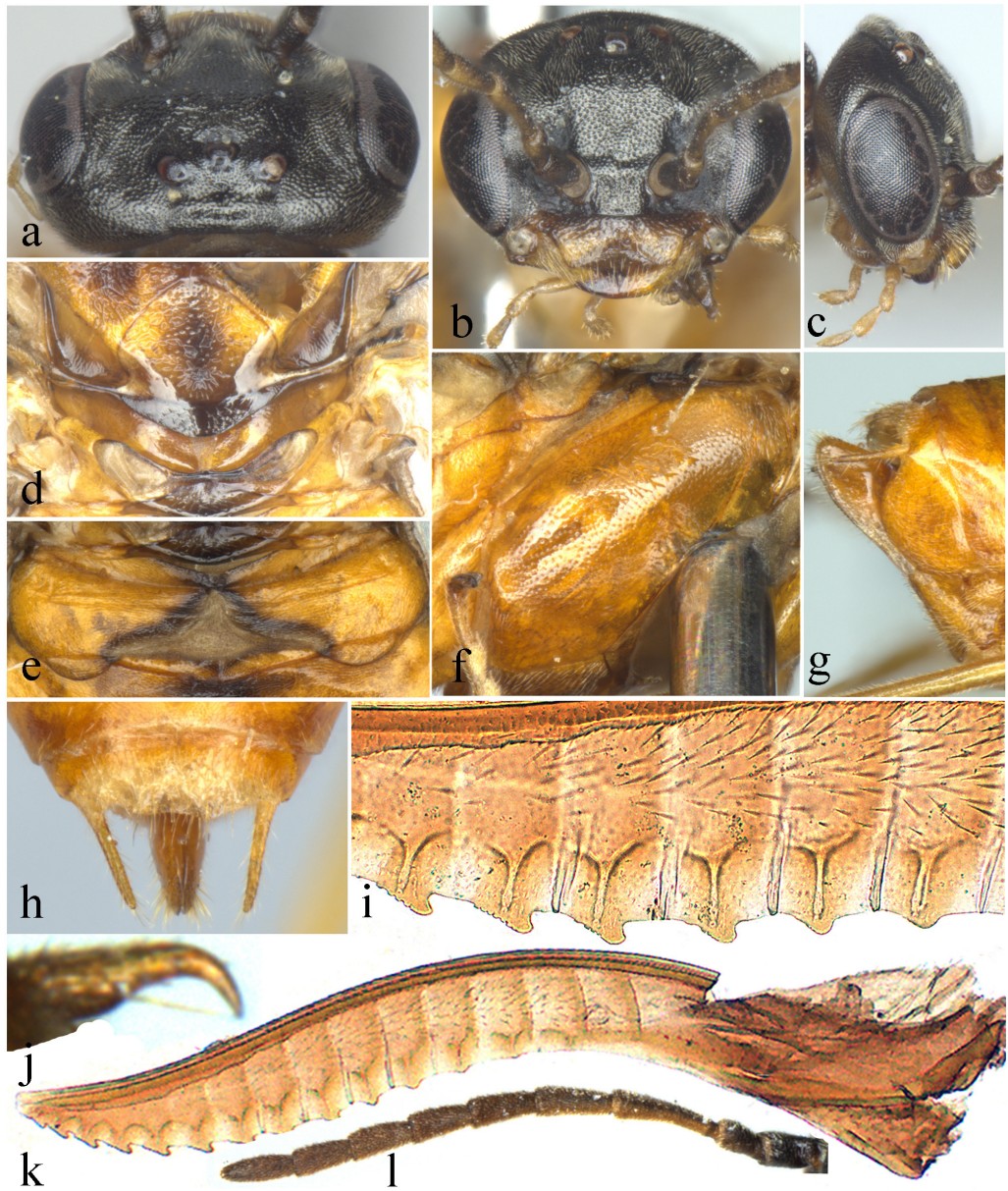

**Figure 2** *Analcellicampa xanthosoma.* (A) Head of female, dorsal view; (B) Head of female, front view; (C) Head of female, lateral view; (C) Scutellum of female; (E) Abdominal tergum 1; (F) Mesopleuron of female, lateral view; (G) Ovipositor sheath of female, lateral view; (H) Cercus of female; (I) Middle serrulae of female; (J) Claw of female; (K) Lancet of female; (L) Antenna of female. Photos: Yaoyao Zhang.

metascutellum, narrow inner margins of first terga and center of abdominal tergites 2–3 dark brown (Fig. 1). Body hairs brown. Wings hyaline, pterostigma and veins pale brown.

Clypeus with large and very shallow punctures, shiny (Fig. 2B); dorsum of head finely punctured with narrow but recognizable interspaces except frons and anterior of temple densely punctured, interspaces linear (Fig. 2A); mesoscutal middle lobe and lateral lobes finely punctured with linear interspaces, most of mesoscutellum with somewhat

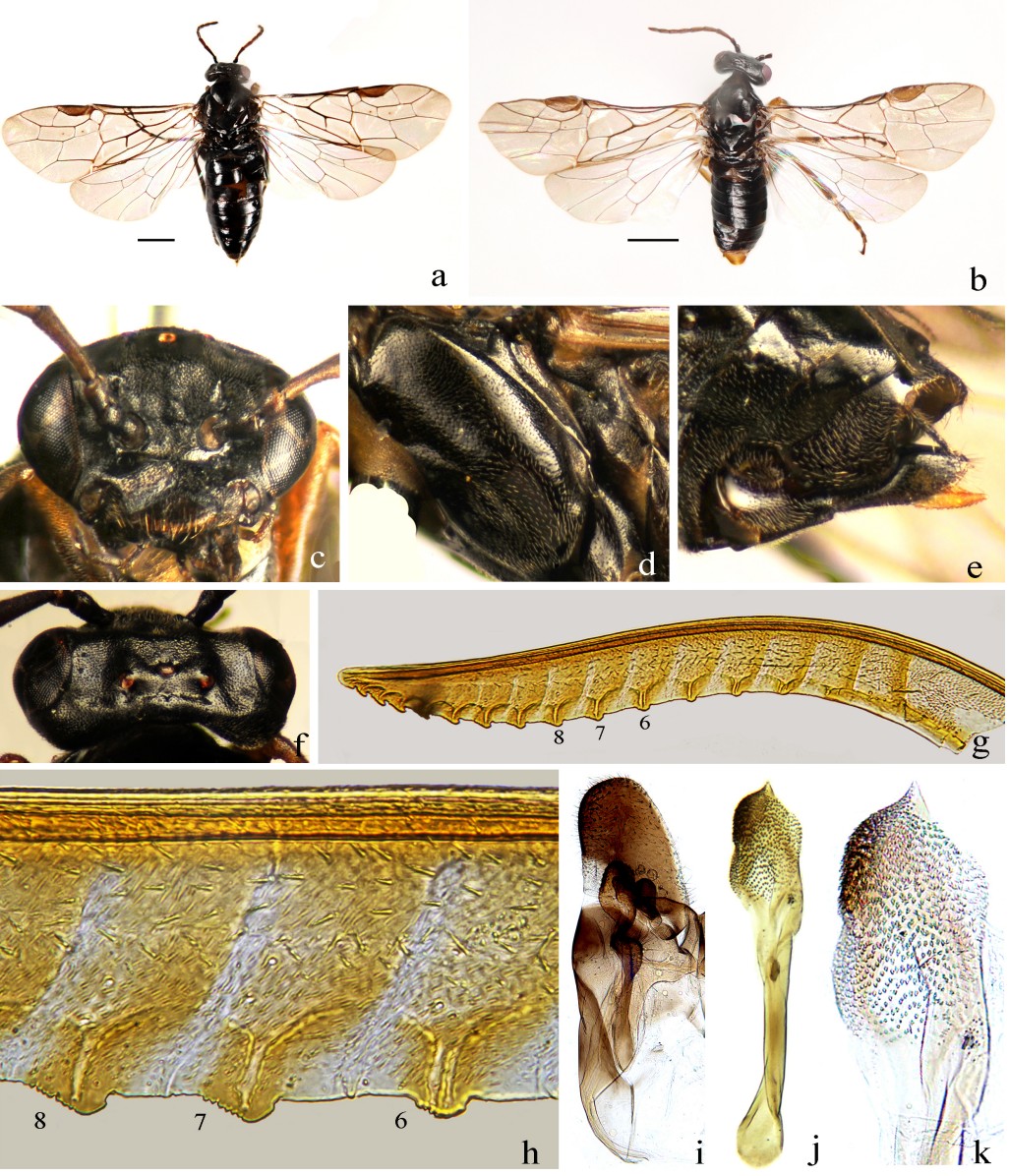

**Figure 3** *Analcellicampa danfengensis* (**Xiao**). (A) Female, dorsal view; (B) Male, dorsal view; (C) Head of female, front view; (D) Mesopleuron, female; (E) Ovipositor sheath, lateral view; (F) Head of female, dorsal view; (G) Lancet; (H) Middle serrulae; (I) Gonoharpe; (J) Penis valve; (K) Valviceps of penis valve. Scale bars = 1 mm. Photos: Meicai Wei.

larger punctures, interspaces distinct and shiny; parapsis largely and lateral corners of mesoscutellum highly smooth, strongly shiny; mesoscutellar appendage and metascutellum distinctly punctured (Fig. 2D), metapostnotum coriaceous; mesepisternum sparsely and finely punctured, interspaces broad and smooth, shiny (Fig. 2F); metapleuron finely punctured with weak microsculptures, weakly shiny; abdominal tergum 1 distinctly microsculptured (Fig. 2E), other tergites weakly coriaceous, shiny.

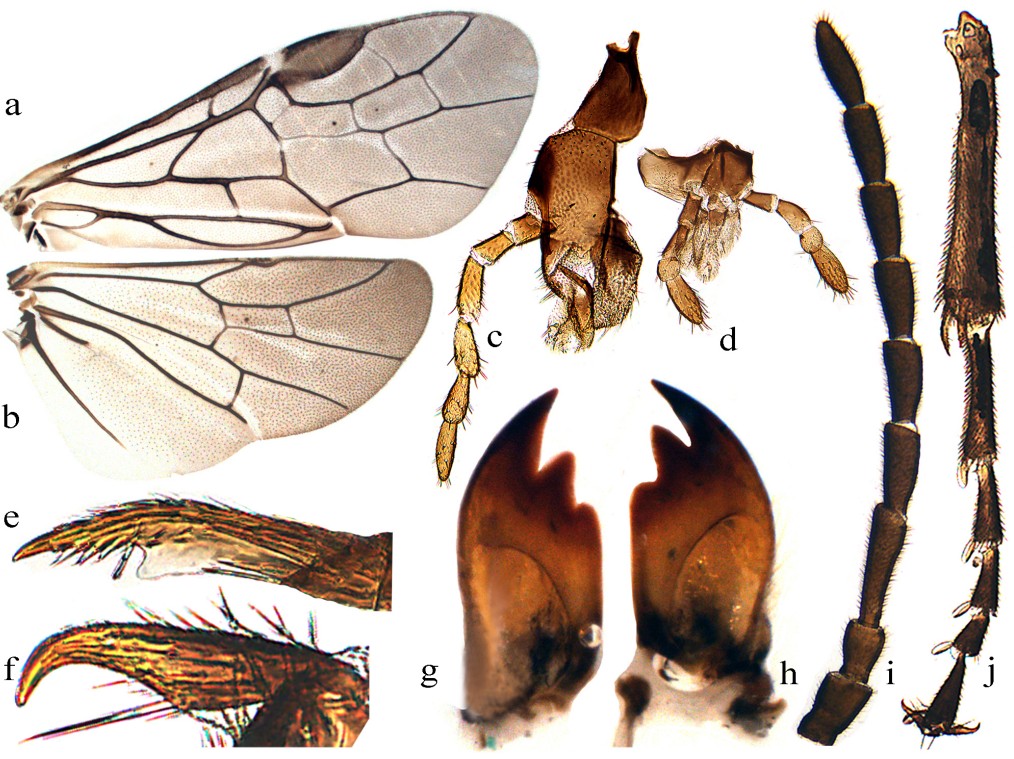

**Figure 4** *Analcellicampa danfengensis* **(Xiao) Female.** (A) Forewing; (B) Hindwing; (C) Maxilla; (D) Labrum; (E) Inner fore tibial spur; (F) Claw; (G) Left mandible; (H) Right mandible; (I) Antenna; (J) Fore tibia and tarsus. Photos: Meicai Wei.

Hairs on dorsum of head and thorax very short, shorter than 0.2 times the diameter of the lateral ocellus. Clypeus triangularly incised to a depth of approximately 0.3 times the length of the clypeus, malar space 0.6 times the diameter of the median ocellus; middle fovea absent, lateral fovea small but distinct; area in front of ocellar triangle flat and broad (Fig. 2B), distance between lower corner of eyes 1.3 times the longest axis of the eye; frons hardly elevated, frontal wall indistinct, frontal basin absent; anterocellar furrow transverse, distinct, interocellar furrow fovea-like, postocellar furrow short and almost in line with interocellar fovea (Fig. 2A); OOL:POL:OCL = 9:8:6; postocellar area approximately 3 times as broad as long, posterior slightly elevated (Fig. 2A); in dorsal view, head strongly narrowed behind eyes, temple approximately half the length of the eye, lateral view as in Fig. 2C; antenna as long as head and thorax together, clearly shorter than vein C of forewing, antennomere 3 1.35 times as long as antennomere 4, antennomere 8 2.2 times as long as broad. Mesoscutellum and appendage as Fig. 2D; mesepisternum flat (Fig. 2F). Hind tibia slightly longer than tarsus, metabasitarsus as long as following 3 tarsomeres together; claw simple (Fig. 2J). Abdominal tergum 1 as in Fig. 2E; ovipositor sheath as long as middle tarsus and 0.8 times as long as hind tibia, apical sheath 1.6 times as long as basal sheath, apex subtriangular in lateral view (Fig. 2G); cercus very long and slender, length approximately 10 times its middle breadth and reaching apex of sheath in dorsal view

(Fig. 2H). Lancet with 14 oblique serrulae and 10 distinct annular sutures, apical 6 annuli without annular suture (Fig. 2K); subapical serrulae each with 7–9 small distal subbasal teeth, proximal subbasal tooth absent (Fig. 2I).

**Male.** Unknown.

**Holotype.** ♀, CSCS16007, China, Hunan, Wugang, Mt. Yunshan, Yunfengge, 110°37′10″E, 26°38′59″N, alt. 1170 m, March 27, 2016, Meicai Wei and Gengyun Niu leg.

**Paratypes.** 1 ♀, data same as the holotype; 1 ♀, CSCS16007, China, Hunan, Wugang, Mt. Yunshan, Yunfengge, 110°37′10″E, 26°38′59″N, alt. 1,170 m, March 27, 2016, Meicai Wei and Gengyun Niu leg, CSCS-Hym-MC0017; 1 ♀, CSCS16007, China, Hunan, Wugang, Mt. Yunshan, Yunfengge, 110°37′10″E, 26°38′59″N, alt. 1170 m, March 27, 2016, Meicai Wei and Gengyun Niu leg, CSCS-Hym-M02046; 1 ♀, CSCS12001, China, Hunan, Wugang, Mt. Yunshan, Dianshita, 110°37.299′E, 26°38.630′N, alt. 1380 m, April 9, 2012, Zejian Li, Zaiyang Pan leg; 4 ♀ ♀, CSCS12001, China, Hunan, Wugang, Mt. Yunshan, Dianshita, 110°37.299′E, 26°38.630′N, alt. 1,380 m, April 9, 2012, Zejian Li, Zaiyang Pan leg, CSCS-Hym-M2018, M2019, M2020, M2043.

**Etymology.** The specific epithet refers to the body color.

**Distribution.** China (Hunan).

**Remarks.** This new species differs from the type species of the genus by the following: thorax, abdomen and legs almost entirely yellow-brown, wings hyaline, the pterostigma and veins pale brown, postocellar area approximately 3 times as broad as long, very long and slender cerci approximately 10 times as long as broad, the ovipositor apical sheath 1.6 times as long as basal sheath and the basal 10 annular sutures of lancet distinct. In *Analcellicampa danfengensis*, the body and legs entirely black in females, wings infuscate toward the apex, pterostigma and veins black-brown, postocellar area approximately 4–4.5 times as broad as long, cerci approximately 5 times as long as broad, the ovipositor apical sheath 1.2 times as long as basal sheath and all annular sutures of lancet indistinct.

***Analcellicampa danfengensis*** (*Xiao, 1994*) **comb. nov.** (**Figs. 3–4**)

*Hoplocampa danfengensis* G. (*Xiao, 1994*): 442–444.

Notes. Body length 4–4.5 mm. Black (Figs. 3A and 3B); antenna black-brown (Fig. 4I), tibiae and base of basitarsi pale brown to dark brown; subgenital plate of male yellow-brown (Fig. 3B). Wings infuscate toward apex, vein C and stigma dark brown to black-brown (Figs. 4A and 4B). Anterior incision of clypeus roundish, shallow; interocellar and postocellar furrows fine and shallow; posterior of postocellar area weakly elevated, approximately 4–4.5 times as broad as long (Fig. 3F). Body finely and densely punctured, abdomen mixed with microsculptures. Antennomere 3 1.3 times as long as antennomere 4 (Fig. 4I). Epicnemial suture fine (Fig. 3D). Cercus slender, approximately 5 times as long as broad (Fig. 3E). Ovipositor apical sheath 1.2 times as long as basal sheath (Fig. 3E); lancet with 17–19 serrulae, annular sutures indistinct (Fig. 3G), middle serrulae distinctly oblique, with indistinct fine subbasal teeth (Fig. 3H). Gonoforcep as shown in Fig. 3I, harpe longer than broad; penis valve as shown in Fig. 3J, apical lobe short and small, valviceps with dense small spines (Fig. 3K).

Distribution. China (Gansu, Shaanxi, Hunan, Zhejiang, and Sichuan).

Host plant: *Cerasus pseudocerasus* (Lindl.).

*Sun & Jiang (1994)* reported on the biology of this species but under the name *Fenusa* sp. The name was defined by Gangrou Xiao in 1994. Later in the same year, Xiao described the species as new to science with the name *Hoplocampa danfengensis*.

## A revised key to the tribes and known genera of Hoplocampinae

1 Antennal flagellum shorter than 2 times head breadth, middle flagellomeres less than 3 times as long as broad; second antennomere clearly longer than broad; forewing with vein 2r usually present (except *Caulocampus*); lancet narrow and long, not distinctly sclerotized, without ctenidium; larvae feeding internally on some species of Angiospermae. Hoplocampini…2

- Length of antennal flagellum more than 2 times head breadth, middle flagellomeres clearly more than 3 times as long as broad or distinctly branched; forewing with vein 2r absent; lancet usually strongly sclerotized with distinct ctenidia; larvae feeding externally on some species of Gymnospermae or Angiospermae…6

2 Forewing with vein 2r absent; larvae boring into leaf petioles or unknown…3

- Forewing with 2r present; larvae tunneling into and feeding within fruits…4

3 Claw short and strongly bent at middle with a large inner tooth; apex of hind anal cell truncate, petiole of anal cell extending from dorsal margin of cell A; larvae boring into leaf petioles. North America…*Caulocampus* (*Rohwer, 1912*)

- Claw slender and not strongly bent at middle, without inner tooth; apex of hind anal cell acute at apex and petiole of anal cell extending from apex of cell A; larvae unknown…*Armenocampus* (*Zinovjev, 2000*)

4 Cell M in hind wing closed; claw with minute inner tooth; penis valve simple or with a long apical filament, without many small spines or a large valvispina, usually with some warts; lancet usually with spiculella, and serrula with several large teeth…*Hoplocampa* Hartig, 1837

- Cell M in hind wing open; claw without inner tooth; penis valve with a distinct subapical valvispina or simple with many small spines; lancet without spiculella, serrula with several minute teeth…5

5 Hind anal cell broadly open with a short 2A vein; penis valve simple without a subapical valvispina but with many small spines; gonocardo quite narrow…*Analcellicampa* Wei & Niu, gen. nov.

- Hind anal cell closed and vein 2A as long as 1A; penis valve with a distinct subapical valvispina, without many small spines; gonocardo very broad…*Monocellicampa* (*Wei, 1998*)

6 Forewing with vein 2r usually present; epicnemium of mesepisternum absent or very narrow and flat; petiole of hind anal cell about as long as vein cu-a; pseudoceps of penis valve with a stout spine, paravalva simple; larvae feeding on leaves of Cupressaceae, Gymnospermae. North America. Susanini…*Susana* (*Rohwer & Middleton, 1932*)

- Forewing with vein 2r always absent; epicnemium broad and flat or narrow and strongly elevated; petiole of hind anal cell much longer than vein cu-a or shorter than cu-a;

pseudoceps of penis valve simple, without a stout spine, paravalva usually with a stout spine (valvispina); larvae feeding on leaves of several families of Angiospermae…7

7 Mesepisternum with a narrow and strongly elevated epicnemium; forewing with vein R+M clearly longer than vein cu-a, first abscissa of vein Rs complete; petiole of hind anal cell shorter than vein cu-a; left mandible in outer view slender in apical 0.7 and strongly enlarged in basal 0.3. Anhoplocampini…8

- Mesepisternum with a broad and flat epicnemium; forewing with vein R +M clearly shorter than vein cu-a, first abscissa of vein Rs absent; petiole of hind anal cell much longer than vein cu-a; left mandible in outer view evenly enlarged toward base. Holarctic. Cladiini…*Cladius* Illiger, 1807

8 Antenna simple, filiform; forewing with vein R shorter than half length of vein Sc; middle petiole of anal cell in forewing approximately as long as vein cu-a; forewing with a distinct transversal macula. China... *Anhoplocampa* (*Wei, 1998*)

- Antennal flagellomeres 3–8 with distinct apical process or long lobe; forewing with vein R clearly longer than vein Sc; middle petiole of anal cell in forewing less than half the length of vein cu-a; forewing without a transversal macula. China…*Zhuangzhoua* Wei et al. 2018.

## Architecture and nucleotide composition of the *A. xanthosoma* mitochondrial genome

The nearly complete mitochondrial genome of *A. xanthosoma* had a length of 15,512 bp and contained 37 genes (22 tRNAs, 13 PCGs, and two rRNAs). The mitogenome sequence was deposited in GenBank under the accession number MH992752. The metadata file of the mitogenome sequence was deposited in GenBank under the SRA accession number SUB4559596. The 37 genes, except for four PCGs (*ND1, ND4, ND4L* and *ND5*), two rRNAs and eight tRNAs, were located on the J-strand (Table 2).

The gene order was the same as the ancestral order except for the IQM cluster (Fig. 5). Accordingly, the causes of mitochondrial genome rearrangements in insects are likely multifactorial, and much additional research is required (*Cameron, 2014*). In the *A. xanthosoma* mitochondrial genome, a total of 239 bp of intergenic spacer sequences were found in 18 locations (except for the A + T-rich region) and varied in size from one to 29 bp. Additionally, a total of 11 overlapping nucleotides were scattered in five locations, with the longest (ATGATAA) located between *ATP8* and *ATP6*.

## Protein-coding genes and codon usage

All PCGs were initiated with ATN as the start codon and ended with a complete termination codon except for the *ND3* and *ND4* genes, which terminated with an incomplete stop codon T (Table 2).

The A + T content, AT-skew and GC-skew are three parameters usually used in investigations of nucleotide composition in mitochondrial genomes (*Hassanin, Léger & Deutsch, 2005*; *Perna & Kocher, 1995*). The *A. xanthosoma* mitochondrial genome was biased toward A and T, with an 80.0% A + T content (Table 3). The AT- and GC-skews were found to be mostly negative in different regions of the *A. xanthosoma* mitochondrial

**Table 2 Mitochondrial genome characteristics of *Analcellicampa xanthosoma*.**

| Gene | Strand | Start | Stop | Length (bp) | Start codon | Stop codon | IGN |
|------|--------|-------|------|-------------|-------------|------------|-----|
| *trnI* | J | 1 | 67 | 67 | | | 5 |
| *ND2* | J | 73 | 1,119 | 1,047 | ATG | TAA | 6 |
| *trnW* | J | 1,126 | 1,195 | 70 | | | −1 |
| *trnC* | N | 1,195 | 1,262 | 68 | | | 0 |
| *trnY* | N | 1,263 | 1,329 | 67 | | | 9 |
| *COX1* | J | 1,339 | 2,877 | 1,539 | ATA | TAA | 29 |
| *trnL2* | J | 2,907 | 2,974 | 68 | | | 1 |
| *COX2* | J | 2,976 | 3,656 | 681 | ATG | TAA | 29 |
| *trnK* | J | 3,686 | 3,756 | 71 | | | −1 |
| *trnD* | J | 3,756 | 3,823 | 68 | | | 0 |
| *ATP8* | J | 3,824 | 3,982 | 159 | ATC | TAA | −7 |
| *ATP6* | J | 3,976 | 4,653 | 678 | ATG | TAA | −1 |
| *COX3* | J | 4,653 | 5,441 | 789 | ATG | TAA | 11 |
| *trnG* | J | 5,453 | 5,519 | 67 | | | 0 |
| *ND3* | J | 5,520 | 5,871 | 352 | ATT | T | 0 |
| *trnA* | J | 5,872 | 5,935 | 64 | | | 14 |
| *trnR* | J | 5,950 | 6,019 | 70 | | | 2 |
| *trnN* | J | 6,022 | 6,088 | 67 | | | 0 |
| *trnS1* | J | 6,089 | 6,156 | 68 | | | 0 |
| *trnE* | J | 6,157 | 6,226 | 70 | | | 4 |
| *trnF* | N | 6,231 | 6,300 | 70 | | | 10 |
| *ND5* | N | 6,311 | 8,035 | 1,725 | ATT | TAA | 0 |
| *trnH* | N | 8,036 | 8,100 | 65 | | | 0 |
| *ND4* | N | 8,101 | 9,445 | 1,345 | ATG | T | 4 |
| *ND4L* | N | 9,450 | 9,749 | 300 | ATG | TAA | 2 |
| *trnT* | J | 9,752 | 9,819 | 68 | | | 0 |
| *trnP* | N | 9,820 | 9,888 | 69 | | | 5 |
| *ND6* | J | 9,894 | 10,409 | 516 | ATG | TAA | −1 |
| *CYTB* | J | 10,409 | 11,545 | 1,137 | ATG | TAA | 5 |
| *trnS2* | J | 11,551 | 11,618 | 68 | | | 19 |
| *ND1* | N | 11,638 | 12,588 | 951 | ATA | TAA | 0 |
| *trnL1* | N | 12,589 | 12,657 | 69 | | | 0 |
| *rrnL* | N | 12,658 | 13,993 | 1,336 | | | 0 |
| *trnV* | N | 13,994 | 14,064 | 71 | | | 0 |
| *rrnS* | N | 14,065 | 14,864 | 800 | | | 33 |
| *trnM* | J | 14,898 | 14,966 | 69 | | | 51 |
| *trnQ* | N | 15,018 | 15,086 | 69 | | | 0 |
| AT-rich region | J | 15,087 | >15,512 | >426 | | | |

**Notes.**
J, major; N, minor; IGN, intergenic nucleotides.
Minus indicates overlapping sequences between adjacent genes.

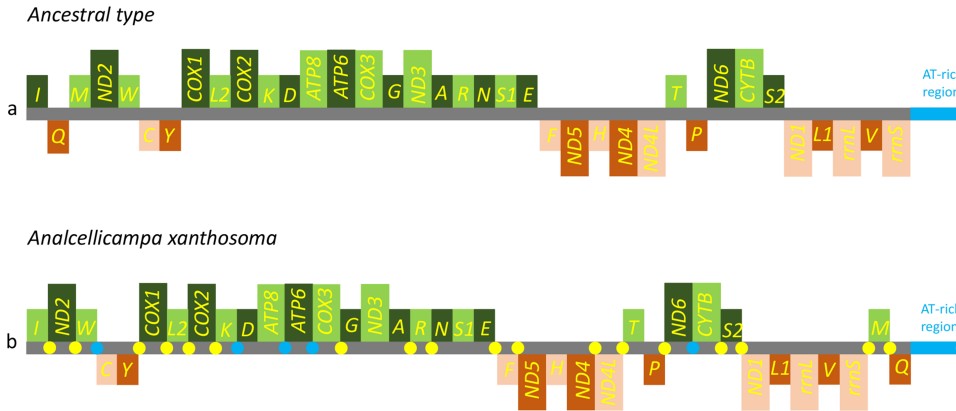

**Figure 5** **Mitochondrial genome organization of *Analcellicampa xanthosoma*.** Mitochondrial genome organization of *A. xanthosoma* with reference to the ancestral type of insect mitochondrial genomes. Genes transcribed from the J- and N-strands are shown in green and orange, respectively. Overlapping and intergenic regions are indicated by yellow and blue circles, respectively. The A+ T-rich region is indicated by blue, and tRNA genes are labeled by their single-letter amino acid code.

genome (Table 3). An investigation of nucleotide skew statistics for the mitochondrial genome of *A. xanthosoma* revealed that PCGs in both strands were T-skewed, whereas the PCGs were C-skewed in the J-strand and G-skewed in the N-strand.

Codon usage of the *A. xanthosoma* mitochondrial genome was shown in Table 4. We observed a relationship between the base composition of codon families and amino acid occurrence, which was assessed by calculating the number of G + C-rich codons (Pro, Ala, Arg, and Gly) and A + T-rich codons (Phe, Ile, Met, Tyr, Asn, and Lys) and then calculating their ratio (*Crozier & Crozier, 1993*). In *A. xanthosoma*, this ratio was 0.16, which was lower than that in *M. pruni* (*Wei, Wu & Liu, 2015*).

After several calculations, multiple observations were made. UUA-Leu had the highest RSCU, with an average value of 3.84. In addition, all codons with an RSCU greater than 2.00 had an A at the third codon position (Table 4). The A content of the third codon position was higher than that of the first codon position and lower than that of second codon positions in PCGs (Table 3). We therefore concluded that the nucleotide composition was closely related to codon usage.

## Transfer RNA genes

The mitochondrial genome of *A. xanthosoma* contains 22 tRNA genes. Fourteen of the 22 tRNA genes were located on the J-strand, while the remaining were encoded by the opposite N-strand. The length of the tRNAs ranged from 64 bp (*trnA*) to 71 bp (*trnK*, *trnV*). All predicted secondary structures of the tRNAs had a common cloverleaf pattern, with the exception of *trnS1* (AGN), where the DHU arm was missing. Predicted secondary structures of the 22 tRNA genes were shown in Fig. 6.

The size of the variable loop and D-loop often determines the overall tRNA length (*Navajas et al., 2002*). In the secondary structure of the tRNAs of the *A. xanthosoma* mitochondrial genome, the DHU arms ranged between 3 and 4 bp in length, the AC arms

**Table 3 Nucleotide composition of *Analcellicampa xanthosoma* mitochondrial genome.**

| Feature | Length (bp) | A% | C% | G% | T% | A + T% | AT-skew | GC-skew |
|---|---|---|---|---|---|---|---|---|
| Whole genome | 15,512 | 42.6 | 11.9 | 7.9 | 37.5 | 80.1 | 0.0637 | −0.2020 |
| Protein coding genes | 11,180 | 34.6 | 10.7 | 10.3 | 44.4 | 79.0 | −0.1241 | −0.0190 |
| First codon position | 3,727 | 34.9 | 10.2 | 12.2 | 43.0 | 77.9 | −0.1040 | 0.0893 |
| Second codon position | 3,727 | 30.6 | 12.5 | 10.5 | 46.0 | 76.6 | −0.2010 | −0.0870 |
| Third codon position | 3,726 | 38.2 | 9.4 | 8.3 | 44.0 | 82.2 | −0.0706 | −0.0621 |
| Protein coding genes-J | 6,871 | 36.9 | 12.9 | 9.4 | 40.8 | 77.7 | −0.0502 | −0.1570 |
| First codon position | 2,291 | 40.3 | 11.3 | 12.3 | 36.0 | 76.3 | 0.0564 | 0.0424 |
| Second codon position | 2,290 | 26.0 | 18.3 | 12.5 | 43.0 | 69.0 | −0.2464 | −0.1883 |
| Third codon position | 2,290 | 44.2 | 9.2 | 3.4 | 43.0 | 87.2 | 0.0138 | −0.4603 |
| Protein coding genes-N | 4,309 | 31.0 | 7.1 | 11.9 | 50.0 | 81.0 | −0.2346 | 0.2526 |
| First codon position | 1,437 | 34.7 | 5.1 | 12.3 | 48.0 | 82.7 | −0.1608 | 0.4138 |
| Second codon position | 1,436 | 23.3 | 11.7 | 14.9 | 50.0 | 73.3 | −0.3643 | 0.1203 |
| Third codon position | 1,436 | 35.0 | 4.7 | 8.4 | 52.0 | 87.0 | −0.1954 | 0.2824 |
| *ATP6* | 678 | 36.1 | 12.8 | 9.1 | 41.9 | 78.0 | −0.0744 | −0.1689 |
| *ATP8* | 159 | 42.1 | 8.2 | 2.5 | 47.2 | 89.3 | −0.0571 | −0.5327 |
| *COX1* | 1,539 | 33.3 | 14.6 | 12.7 | 39.4 | 72.7 | −0.0839 | −0.0696 |
| *COX2* | 681 | 40.4 | 12.8 | 9.3 | 37.6 | 78.0 | 0.0359 | −0.1584 |
| *COX3* | 789 | 34.0 | 15.6 | 11.9 | 38.5 | 72.5 | −0.0621 | −0.1345 |
| *CYTB* | 1,137 | 33.9 | 14.1 | 10.9 | 41.1 | 75.0 | −0.0960 | −0.1280 |
| *ND1* | 951 | 49.1 | 12.8 | 8.4 | 29.7 | 78.8 | 0.2462 | −0.2075 |
| *ND2* | 1,047 | 42.7 | 9.6 | 4.9 | 42.8 | 85.5 | −0.0012 | −0.3241 |
| *ND3* | 352 | 36.1 | 11.4 | 8.5 | 44.0 | 80.1 | −0.0986 | −0.1457 |
| *ND4* | 1,345 | 50.3 | 12.1 | 7.0 | 30.6 | 80.9 | 0.2435 | −0.2670 |
| *ND4L* | 300 | 53.3 | 13.3 | 1.7 | 31.7 | 85.0 | 0.2541 | −0.7733 |
| *ND5* | 1,725 | 49.7 | 10.9 | 7.5 | 31.9 | 81.6 | 0.2181 | −0.1848 |
| *ND6* | 516 | 41.5 | 10.3 | 5.4 | 42.8 | 84.3 | −0.0154 | −0.3121 |
| *rrnL* | 1,336 | 45.4 | 10.9 | 5.7 | 38.0 | 83.4 | 0.0887 | −0.3133 |
| *rrnS* | 800 | 43.6 | 11.5 | 6.1 | 38.8 | 82.4 | 0.0583 | −0.3068 |

ranged between 4 and 5 bp in length, and the T ΨC arms varied from 3 to 5 bp in length. The length of the amino acid acceptor (AA) stem was conserved at 7 bp in all of the tRNA genes. The anticodon (AC) loops were usually 7 bp in length. The length of the variable loops was less consistent, ranging from 4 to 7 bp.

The variable loops in *trnD, trnF, trnG, trnH* and *trnT* were completely conserved between *A. xanthosoma* and *M. pruni* and differed from those in *Allantus luctifer*, *A. rufocephalus, Tenthredo tienmushana* and *Birmella discoidalisa*. The AC loop in *trnC*, T ΨC loop in *trnS2*, T ΨC arm in *trnY* and AC arm *in trnV* were completely conserved in *A. luctifer, A. rufocephalus, T. tienmushana, A. xanthosoma* and *M. pruni*, while those in *B. discoidalisa* were different. The base between the DHU arm and AC arm in *trnS2* was A in *A. xanthosoma, M. pruni*, and *B. discoidalisa*, while the base was U in *A. luctifer*, *A. rufocephalus* and *T. tienmushana*.

**Table 4  Codon usage of PCGs in mitochondrial genome of *Analcellicampa xanthosoma*.**

| Amino acid | Codon | NO. | RSCU | Amino acid | Codon | NO. | RSCU |
|---|---|---|---|---|---|---|---|
| Phe | UUU | 407 | 1.66 | Tyr | UAU | 200 | 1.69 |
|  | UUC | 82 | 0.34 |  | UAC | 37 | 0.31 |
| Leu | UUA | 308 | 3.84 | End | UAA | 129 | 1.39 |
|  | UUG | 43 | 0.54 |  | UAG | 57 | 0.61 |
| Leu | CUU | 55 | 0.69 | His | CAU | 64 | 1.62 |
|  | CUC | 9 | 0.11 |  | CAC | 15 | 0.38 |
|  | CUA | 53 | 0.66 | Gln | CAA | 58 | 1.81 |
|  | CUG | 13 | 0.16 |  | CAG | 6 | 0.19 |
| Ile | AUU | 352 | 1.73 | Asn | AAU | 227 | 1.77 |
|  | AUC | 55 | 0.27 |  | AAC | 30 | 0.23 |
| Met | AUA | 197 | 1.66 | Lys | AAA | 126 | 1.58 |
|  | AUG | 40 | 0.34 |  | AAG | 33 | 0.42 |
| Val | GUU | 38 | 1.33 | Asp | GAU | 67 | 1.72 |
|  | GUC | 12 | 0.42 |  | GAC | 11 | 0.28 |
|  | GUA | 53 | 1.86 | Glu | GAA | 60 | 1.48 |
|  | GUG | 11 | 0.39 |  | GAG | 21 | 0.52 |
| Ser | UCU | 55 | 1.48 | Cys | UGU | 24 | 1.17 |
|  | UCC | 24 | 0.65 |  | UGC | 17 | 0.83 |
|  | UCA | 102 | 2.75 | Trp | UGA | 73 | 1.47 |
|  | UCG | 8 | 0.22 |  | UGG | 26 | 0.53 |
| Pro | CCU | 31 | 1.51 | Arg | CGU | 4 | 0.67 |
|  | CCC | 14 | 0.68 |  | CGC | 1 | 0.17 |
|  | CCA | 36 | 1.76 |  | CGA | 17 | 2.83 |
|  | CCG | 1 | 0.05 |  | CGG | 2 | 0.33 |
| Thr | ACU | 52 | 1.58 | Ser | AGU | 21 | 0.57 |
|  | ACC | 15 | 0.45 |  | AGC | 13 | 0.35 |
|  | ACA | 61 | 1.85 |  | AGA | 46 | 1.24 |
|  | ACG | 4 | 0.12 |  | AGG | 28 | 0.75 |
| Ala | GCU | 24 | 1.50 | Gly | GGU | 20 | 0.68 |
|  | GCC | 8 | 0.50 |  | GGC | 7 | 0.24 |
|  | GCA | 31 | 1.94 |  | GGA | 74 | 2.51 |
|  | GCG | 1 | 0.06 |  | GGG | 17 | 0.58 |

A total of 21 unmatched base pairs were scattered throughout the 22 tRNA genes, including 13 pairs in the AA stems, five pairs in the DHU stems and three pairs in the T ΨC stems. Fourteen of these unmatched base pairs were G-U pairs, which formed a stable hydrogen-bonded pair. The remaining unmatched pairs were U-U (6) and A-A (1) mismatches. The phenomenon of aberrant mismatches, loops, or extremely short arms for tRNA is common in metazoan mitochondrial genomes (*Wolstenholme, 1992*). Although whether the aberrant tRNAs have lost their respective functions is still unknown, such loss can be corrected by post-transcriptional RNA-editing processes (*Lavrov, 2000*; *Masta & Boore, 2004*).

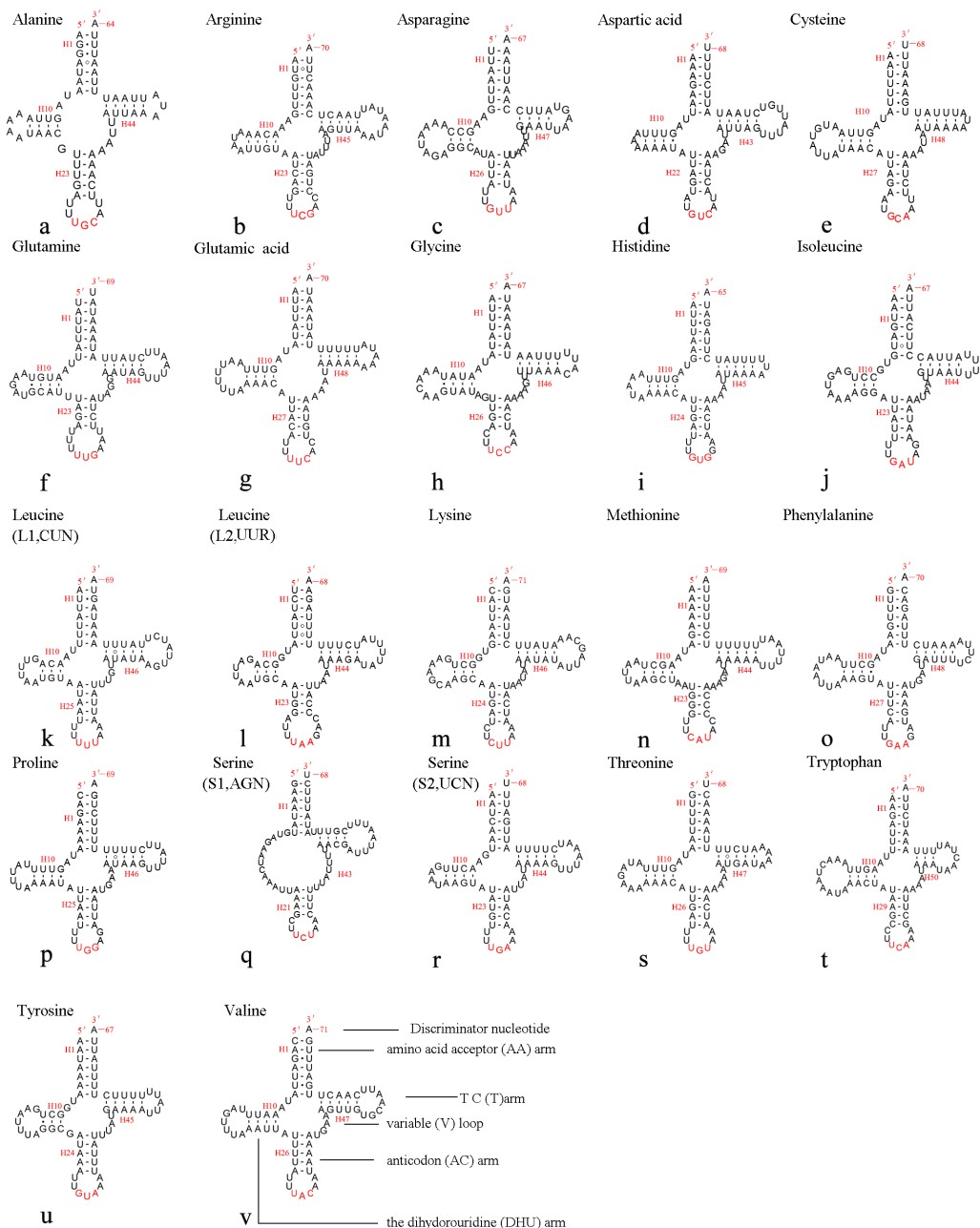

**Figure 6** **Predicted secondary structures of 22 tRNA genes of *A. xanthosoma*.** Dashes indicate Watson-Crick base pairs, and dots indicate G-U base pairing.

## Ribosomal RNA genes

The *rrnL* gene was 1,336 bp in length with an 83.4% A + T content, while *rrnS* was 800 bp in length with an 82.4% A + T content (Table 2). In addition, *rrnL* was located between *trnL1* and *trnV*; *rrnS* was located downstream of *trnV*. The secondary structures of *rrnL* and *rrnS* are shown in Figs. 7 and 8.

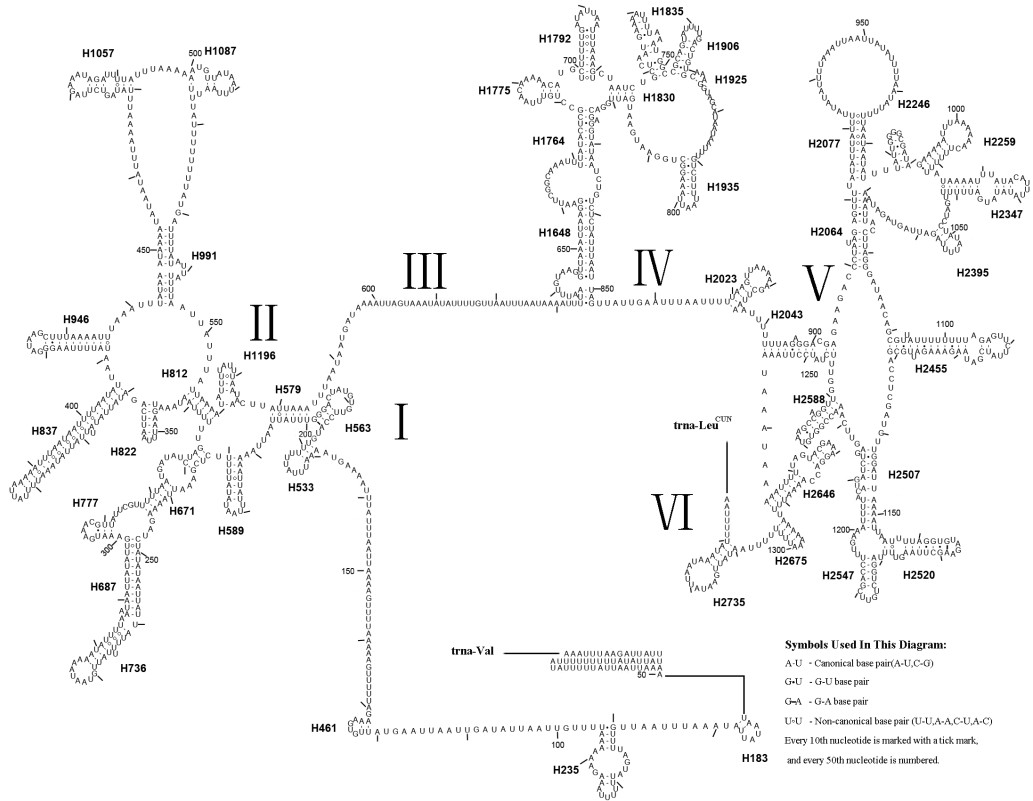

**Figure 7** **Predicted *rrnL* secondary structure in the *A. xanthosoma* mitochondrial genome.** The numbering of helices follows *Gillespie et al. (2006)*. Roman numerals refer to domain names.

Forty-four helices belonging to six domains were identified in the *rrnL* secondary structure of *A. xanthosoma*, similar to the findings for *A. mellifera* (*Gillespie et al., 2006*). In *rrnL*, domains IV and V were more conserved within the Tenthredinidae family. Eight helices (H563, H1775, H1830, H1925, H2023, H2043, H2547 and H2588) were highly conserved. Furthermore, some helices (H837, H991, H1196 and H2347) were highly variable in terms of their sequence and secondary structure compared with other insects (*Gillespie et al., 2006*; *Du et al., 2018*; *Niu et al., 2019*; *Castro & Dowton, 2005*; *Dowton et al., 2009*). Compared with *A. mellifera*, which harbored an A at position 75 (H235), an A at position 238 (H671) and an A at position 733 in H1835, *A. luctifer*, *A. rufocephalus* and *T. tienmushana* harbored a U at each site. H736 and H1764 were completely conserved in helices between *A. xanthosoma* and *M. pruni* but variable in *A. luctifer, A. rufocephalus, T. tienmushana* and *B. discoidalisa.* H687 was highly conserved in helices between *A. xanthosoma* and *M. pruni*, with only one change (an A at position 255 changed to a G). The H777 in *B. discoidalisa* was different from that in *A. luctifer, A. rufocephalus, T. tienmushana*, *A. xanthosoma* and *M. pruni.* Additionally, a U at position 342 was changed to a G in *B. discoidalisa,* and a U at position 510 in H1087 was changed to a C in *B. discoidalisa.* H822 was completely conserved in *A. luctifer, A. rufocephalus, T. tienmushana* and *B. discoidalisa,* while the G-C was replaced by a Watson-Crick A-U

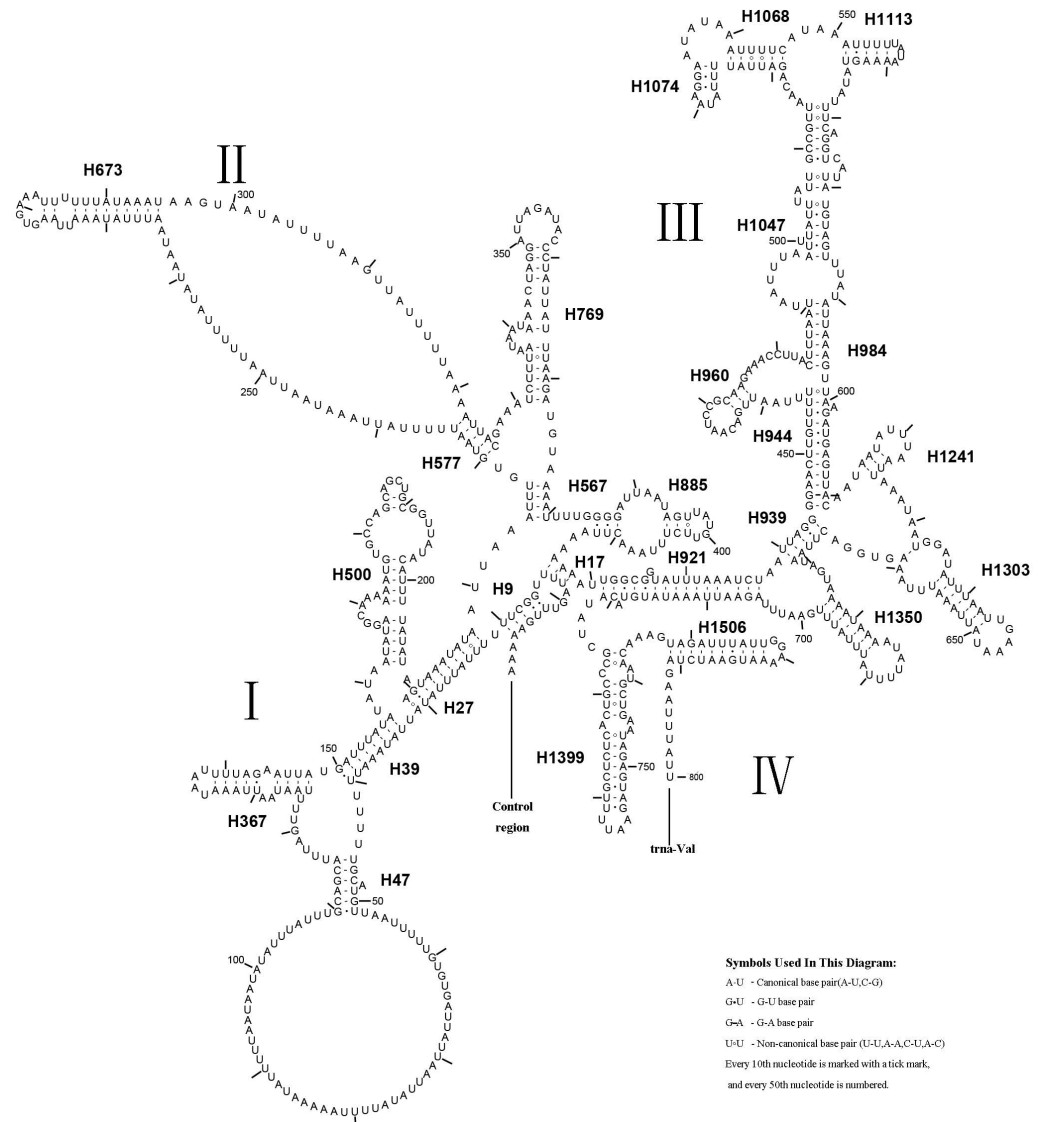

**Figure 8  Predicted *rrnS* secondary structure in the *A. xanthosoma* mitochondrial genome.** The numbering of helices follows *Gillespie et al. (2006)*. Roman numerals refer to domain names.

pair at positions 347-357. H1057 was conserved between *A. xanthosoma* and *M. pruni*, while it was variable in *A. luctifer, A. rufocephalus, T. tienmushana* and *B. discoidalisa.*

*A. xanthosoma* also expressed *rrnS* with 26 helices in four domains, consistent with the pattern observed in Zygaenoidea (*Niehuis, Naumann & Misof, 2006*). Specifically, H47 was variable among the different hymenopteran species, having a large loop. The loop size was variable and determined by the overall *rrnS* length. The H769 and H1399 were more similar to those of *A. mellifera* characterized by *Gillespie et al. (2006)* than to other helices. In *rrnS*, domain III was more conserved within Tenthredinidae than domains I, II, and VI. The loop between H39 and H47, the loop between H984 and H1047 at positions

497–491, as well as the H367 loop were completely conserved between *A. xanthosoma* and *M. pruni* but variable among *A. luctifer, A. rufocephalus, T. tienmushana* and *B. discoidalisa*. Compared with *A. xanthosoma* and *M. pruni*, a C at position 169 was changed to a U, a U at position 461 was changed to a A and a U at position 658 was changed to a G in *A. luctifer, A. rufocephalus, T. tienmushana* and *B. discoidalisa*. Similarly, a C-G pair was replaced by a C-G pair at positions 510–573 in *A. luctifer, A. rufocephalus* and *T. tienmushana*.

The secondary structures of *rrnL* and *rrnS* agreed with the BI and ML analyses that *A. xanthosoma* + *M. pruni* is a sister group to (((*A. luctifer* + *A. rufocephalus*) *T. tienmushana*) *B. discoidalisa*).

### Phylogenetic relationships

To investigate the phylogenetic relationships within Symphyta, we analyzed 27 symphytan and two apocritan mitochondrial genomes. Genomes from four species were also used as outgroups (Mecoptera, Diptera, Megaloptera, and Coleoptera) (Table 1). The 27 species of Symphyta represented eight families: Tenthredinidae (*Wei, Niu & Du, 2014*; *Wei, Wu & Liu, 2015*; *Song et al., 2015*; *Song et al., 2016*; GY Niu, 2017, unpublished data), Cimbicidae (*Song et al., 2016*; *Doğan & Korkmaz, 2017*; YC Yan, 2019, unpublished data), Pergidae (*Castro & Dowton, 2005*), Orussidae (*Dowton et al., 2009*), Cephidae (*Dowton et al., 2009*; *Korkmaz et al., 2015*; *Korkmaz et al., 2016*; *Korkmaz et al., 2017*; *Korkmaz et al., 2018*), Argidae (*Du et al., 2018*), Megalodontesidae, and Pamphiliidae (*Niu et al., 2019*).

Phylogenetic relationships within the suborder Symphyta were reconstructed using both BI and ML analyses (Fig. 9). They both grouped *A. xanthosoma* with *M. pruni* and revealed that *A. xanthosoma* + *M. pruni* is a sister group to (((*A. luctifer* + *A. rufocephalus*) *T. tienmushana*) *B. discoidalisa*) and that Tenthredinidae forms a sister group with Cimbicidae. The systematic position of *Analcellicampa* + *Monocellicampa* within Tenthredinidae (=Tenthredinoidea sensu *Wei & Nie, 1998*) agreed with the system of *Wei & Nie (1998)*. Although there were some differences in our results between the two analytical methods, our findings largely agreed with traditional morphological classifications and recent molecular studies. Additionally, we demonstrated that mitochondrial genome sequences can be used to solve phylogenetic relationships at different taxonomic levels within Symphyta.

However, the relationship between *Analcellicampa* + *Monocellicampa* and other taxa of Nematinae needs further study as no mitochondrial genome has been sequenced for any member of Susanini, Cladiini or Nematini.

## CONCLUSIONS

*Analcellicampa* is a peculiar new genus of Hoplocampinae and is closely allied to *Monocellicampa* Wei. *Analcellicampa* **gen. nov.** differs from the latter by the following characteristics: anal cell of the hind wing broadly opened at apex, the epicnemium large with a distinct epicnemial suture, the third tooth of the mandibles obtuse, the inner tibial spur of the foreleg simple with a membranous lobe far from the apex and the valviceps of the penis valve without a stout valvispina but with many small surface teeth. *A. xanthosoma* **sp. nov.** differs from *A. danfengensis* (*Xiao, 1994*) **comb. nov.** by the following: the thorax,

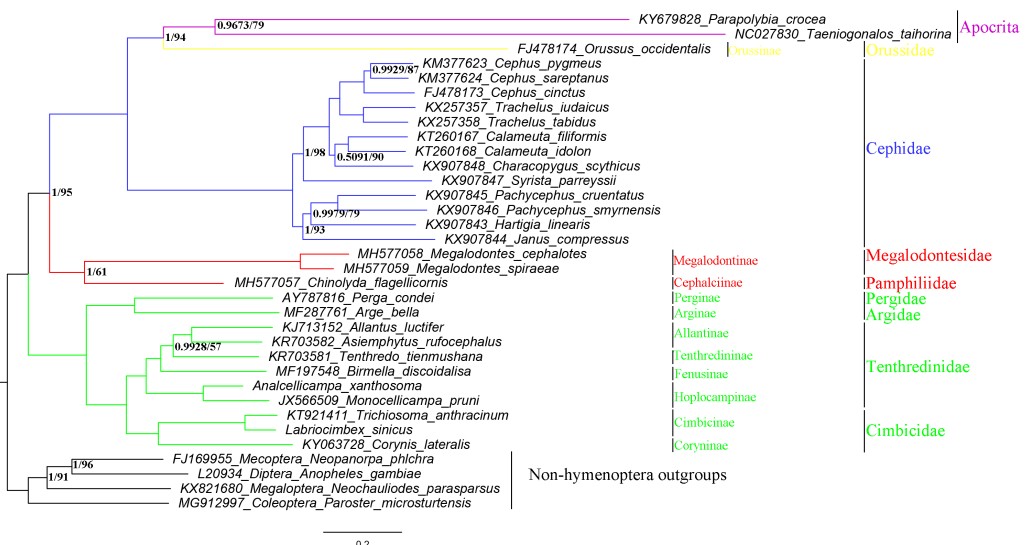

**Figure 9** **Symphytan phylogenetic tree constructed with BI and ML approaches using a mitogenome dataset including 15 individual genes (13 PCGs and two rRNAs).** Both analyses produced the same tree topology. Support values lower than 100% in the ML analysis and 1.0 in the BI analysis were shown.

abdomen and legs almost entirely yellow-brown, the wings hyaline with the pterostigma and veins pale brown, the postocellar area approximately 3 times as broad as long, very long and slender cerci approximately 10 times as long as broad, the ovipositor apical sheath 1.6 times as long as the basal sheath and the basal 10 annular sutures of the lancet distinct.

The nearly complete mitochondrial genome of *A. xanthosoma* was obtained and found to have a length of 15,512 bp and 37 genes (22 tRNAs, 13 PCGs, and two rRNAs). The gene order was the same as its ancestral type. The secondary structures of 22 tRNAs and two rRNAs resembled those of Symphyta, but some helices (H837, H991, H1196 and H2347) were highly variable in *rrnL*. The secondary structure of *rrnL* remains to be studied. Finally, phylogenetic reconstruction based on mitochondrial genomes (13 PCGs and two rRNAs) revealed similarly high support (100%) in both BI and ML analyses, with the result that *A. xanthosoma* was sister to *M. pruni*. We suggest that *A. xanthosoma* as well as *M. pruni* belongs to the tribe Hoplocampini of Hoplocampinae based on adult morphology and molecular data from the mitochondrial genome.

## ACKNOWLEDGEMENTS

The members of the Lab of Insect Systematics and Evolutionary Biology (LISEB) from Central South University of Forestry and Technology are thanked for their contributions to laboratory work. We thank all the reviewers for their comments.

### Funding

The research was supported by the National Natural Science Foundation of China (Nos. 31501885 and 31672344). The funders had no role in study design, data collection and analysis, decision to publish, or preparation of the manuscript.

### Grant Disclosures

The following grant information was disclosed by the authors:
National Natural Science Foundation of China: 31501885, 31672344.

### Competing Interests

The authors declare there are no competing interests.

### Author Contributions

- Gengyun Niu conceived and designed the experiments, analyzed the data, prepared figures and/or tables, approved the final draft, submit to Zoobank.
- Yaoyao Zhang performed the experiments, analyzed the data, prepared figures and/or tables, submit to Genbank.
- Zhenyi Li analyzed the data.
- Meicai Wei contributed reagents/materials/analysis tools, authored or reviewed drafts of the paper.

### DNA Deposition

The following information was supplied regarding the deposition of DNA sequences:
The sequence described here are accessible via GenBank accession number MH992752.

### Data Availability

Niu, Gengyun; Wei, Meicai; Zhang, Yaoyao; Li, Zhenyi (2019): A systematic study of Hoplocampinae. figshare. Fileset. https://doi.org/10.6084/m9.figshare.7121090.v1.
Zhang, Yaoyao (2019): A systematic study of Hoplocampinae. figshare. Fileset. https://figshare.com/projects/A_systematic_study_of_Hoplocampinae/60707.

### New Species Registration

The following information was supplied regarding the registration of a newly described species:
Publication LSID:
urn:lsid:zoobank.org:pub:8093814F-7A3A-460B-A590-4EF928A280E1
Analcellicampa gen. nov. LSID: urn:lsid:zoobank.org:act:FFD0BF94-EF5B-4BE1-8943-72A766426E9A
Analcellicampa xanthosoma sp. nov. LSID: urn:lsid:zoobank.org:act:4EB9F310-6479-4363-A481-81ECF9EF18B6

## Supplemental Information

Supplemental information for this article can be found online at http://dx.doi.org/10.7717/peerj.6866#supplemental-information.

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
