# Peer review of "Characterization of the mitochondrial genome of Analcellicampa xanthosoma gen. et sp. nov. (Hymenoptera: Tenthredinidae)"

_PeerJ, doi:10.7717/peerj.6866_

## Round 0.1 · original submission · Major Revisions

Dear Dr. Niu and colleagues:

Thanks for submitting your manuscript to PeerJ. I have now received two independent reviews of your work, and as you will see, the reviewers raised substantial concerns about the research. Neither reviewer is at all optimistic with the current version of your manuscript. However, if you choose to do so, you may address the issues raised by both reviewers and submit a revision. I cannot guarantee that these reviewers will be willing to evaluate your work again, but a resubmission will again be reviewed.

IMPORTANTLY, please ensure that an English expert has edited your revised manuscript for content and clarity. Also, please ensure that ALL files can be accessed and opened with usual text processors or similar software. Finally, please access the marked-up version of your manuscript kindly provided by reviewer 2, as this outlines many areas for improvement to botht he writing and content.

Accordingly, I am recommending that you revise your manuscript, taking into account all of the issues raised by the reviewers. I look forward to seeing your revision, and thanks again for submitting your work to PeerJ.

Good luck with your revision,

-joe

Reviewer 1 ·

Basic reporting

An extensive language revision of the manuscript is necessary, both in terms of grammar and style of the text (e.g. "clads" [clades] (line 80), "Analcellicampa gen. nov. and Monocellicampa Wei is undoubtedly a member of Hoplocampini" (lines 90-91) [should be given in plural], "according to the manufacture’s protocols" [manufacturer’s protocol] (lines 123-124), "Host plat" [plant] and "Larvae are borer" [borers] (line 223), "epicnemium vestigiel" [vestigial] (line 231), "larvae boring leaf petiole or known" [unknown] (line 324), "which were usually used" (line 387), "condon usage" [codon usage] (line 409), "because of the present" [presence] (line 413), etc.). I therefore strongly recommend the language of the paper to be checked by a native English speaker. Moreover, all genus and species names must be given in italics (please see the first paragraph of the Conclusions section, lines 474-480, as well as the reference list), and some hyphens apparently coming from previous versions of the text (see e.g. lines 389, 400, 405 and 460) should also be removed.

Not all files can be downloaded from the location provided by the authors. I personally could not download figures 2, 6 and 7.

Some sources cited in the reference list, e.g. those by Ashmead, P. Cameron, Dierckxsens et al., Konow, Malaise, Nawrocki, Rohwer and Middleton, Ross, must be carefully checked for both correctness and completeness.

In addition, most files in the Raw data section cannot be opened with usual text processors or similar software.

Experimental design

No comment

Validity of the findings

No comment

Additional comments

The authors list the names of four outgroups used in the analysis (lines 452-453), but I would replace the latter one, Adephaga, to Coleoptera, since the other three represent various insect orders.

Overall, I believe that inaccurate submission of the manuscript (please see above} effectively prevents its further consideration by the Editorial Board of PeerJ.

Reviewer 2 ·

Basic reporting

This manuscript presents the mitochondrial genome of a new species from a new described genus of Analcellicampa with a phylogenetic analysis of Symphyta based on the previously known mitogenome sequences. The authors report general features of this genome and emphasise that this mitochondrial genome shares similar characteristics with the ancestral mitochondrial genome. The phylogenetic tree recovered Analcellicampa and Monocellicampa as sister group within the family of Tenthredinidae. However, the manuscript must be reviewed by a native speaker (or a professional corrector) in order to improve the flow and clarity of text. I have tried to correct as many English mistakes as possible but unfortunately there are far too many and therefore it needs to be checked very thoroughly. Apart from the problem with the English, I have highlighted also other problems on the manuscript that should be solved or explained better. Moreover, it is essential to draw an aim paragraph in the last part of the introduction. Authors should also revise the references both in text and references list adding some missing references. There are also questions regarding the presentation of the results that the authors will have to address.

Experimental design

I think "unlinked branch lengths” should be used in the analysis with PartitionFinder. Moreover, this “linked branch lengths” preference may have led to inaccurate results of phylogenetic analyses. When considering the trees, I clearly realised this case from misplacement of pamphiloid species. They are not clustered together as a monophyletic group.

Validity of the findings

Although the findings have been discussed and supported by relevant literature sufficiently, the main problem is linguistic throughout the MS. It is not clearly presented and needs to be improved.

Additional comments

Here, the manuscript presents the characterisation of the mitochondrial genome of Analcellicampa xanthosoma for the first time. Then the authors aimed to compare the various genomic features against other reported symphytan species. The manuscript presents a newly built phylogenetic tree using the known symphytan mitochondrial genomes to verify the phylogenetic position of the species. All changes are marked in the MS and some major/minor changes or comments are also listed below:
• Lines 85-88: “It seems that the branch of (Susana + Hoplocampa + Cladius) of Nematinae s. lat. is probably a monophyletic group, though the systematic position and the in groups of the branch are uncertain: the position of Monocellicampa, Craterocercus and Moricella is a question.” What means of “in groups of the branch”?
• Lines 135-136: This sentence should be moved to the beginning of the paragraph.
• Line 149-154: “The secondary structures of the rrnS and rrnL were partitioned into 4 areas and 6 areas entirely. The structures were artificially transformed to the relative secondary structure of Italian honey bee. The structural adjustment of remainders was predicted by specific structure models, which were built in SSU-ALIGN (Nawrocki, 2009) with multiple sequence alignments from MARNA (Siebert & Backofen, 2005), in view of the primary sequence and the secondary structure of above-mentioned partial species as well.” It is unclear and it would be better to rewrite considering the related literature.
• Lines 434-435: “The rrnL and the rrnS that accounted for about 9% and 5% were essential for the translation of messenger RNAs into mitochondrial proteins.” What means of this sentence? I think this is not necessary.
• Lines 445-446: “The number of the loop was determined by its length.” What do you mean?
• The authors should rearrange the order of Tables, which I highlighted at the MS.
• The figures 9 and 10 should be coloured and presented using only species name (removing accession numbers) demonstrating their family and superfamily names.

Annotated reviews are not available for download in order to protect the identity of reviewers who chose to remain anonymous.

---

## Round 0.2 · Major Revisions

Dear Dr. Niu and colleagues:

Thanks for re-submitting your manuscript to PeerJ. I have now received two independent reviews of your work (both from the previous reviewers), and as you will see, the reviewers still have some concerns about the research. However, both reviewers are much more optimistic with this version of your manuscript! Accordingly, please address these issues raised by both reviewers and submit another revision. I am sure this second revision will bring you close to acceptance for publication.

IMPORTANTLY, please ensure that an English expert has edited your revised manuscript for content and clarity. Reviewer 2 has kindly provided a marked-up version of your revision, but you should still seek out an English expert.

I look forward to seeing your revision, and thanks again for submitting your work to PeerJ.

Good luck with your revision,

-joe

Reviewer 1 ·

Basic reporting

Dear Colleagues,
I have carefully read the revised version of the manuscript by Niu et al. entitled "Characterization of the mitochondrial genome of Analcellicampa xanthosoma gen. et sp. nov. (Hymenoptera: Tenthredinidae)". As I wrote you earlier, the paper apparently contains important information on the new sawfly genus and species together with its mitochondrial genome, and therefore I believe that it could be published in PeerJ. In addition, substantial changes have been done to the recent version of the manuscript to meet some of the reviewers’ suggestions. However, certain issues related to the paper still remain unresolved.
Specifically, an extensive language revision of some parts of the manuscript, especially of the abstract and introduction, is necessary. During the previous round of the reviewing procedure, I strongly recommended a revision of that kind by a native English speaker. Nevertheless, it seems that the paper still contains many language errors, although the authors state in their rebuttal letter that they "have carefully proof-reading the manuscript to minimize the errors" (by the way, the latter phrase is grammatically incorrect as well). Moreover, section "Etymology" in the description of the genus Analcellicampa is obviously duplicated (see lines 231-232 and 233-235). Furthermore, ALL genus and species names given in the reference list must be given in italics. In addition, a few sources cited in the list must be carefully checked for both correctness and completeness (e.g. authors’ surnames in line 528 should be spelled out).

Experimental design

No comment

Validity of the findings

No comment

Additional comments

Overall, I believe that the manuscript can be published in PeerJ only after a careful major revision of all sections of the text and figures.

Reviewer 2 ·

Basic reporting

I am happy to see the revised version of the manuscript about the mitogenome features and taxonomic position of A. xanthosoma. I am also glad to see that newly performed phylogenetic analyses were included. I feel that the text improved but I have highlighted some English mistakes and some modifications on the manuscript attached. Also, the manuscript must still be reviewed by a native speaker (or a professional corrector) in order to improve the flow and clarity of text and to avoid unnecessary repetitions. Moreover, as I stated before, it is essential to draw an aim paragraph in the last part of the introduction. There are also questions regarding the presentation of the results. Figure 6 seems to be problematic. I couldn’t see some tRNAs in the figure. Figures 9 and 10 are on phylogenetic trees obtained from ML and BI methods. Both trees have same topology and therefore could be presented as a single tree highlighting the supports values of each methods. Posterior probabilities should be shown as decimal. Also, Symphyta is a paraphyletic group and Orussoidea is thought to be a sister group with Apocrita. So I think it’s better to rotate the clade including "Apocrita + Orussoidea + Cephoidea + Pamphilioidea" in the phylogenetic tree to see the relationship between Orussoidea and Apocrita.

Experimental design

no comment.

Validity of the findings

no comment.

Additional comments

no comment.

Annotated reviews are not available for download in order to protect the identity of reviewers who chose to remain anonymous.

---

## Round 0.3 · Major Revisions

Dear Dr. Niu and colleagues:

Thanks for once again re-submitting your manuscript to PeerJ. I have now received two independent reviews of your work (both from the previous reviewers), and as you will see, the reviewers still have some concerns about the research. Note that reviewer 1 was very unhappy about your failure to correct the English and grammar in the resubmission, and now we have lost that reviewer moving forward. However, reviewer 2 has remained dedicated to the review process and has very graciously once again helped improve your manuscript. Accordingly, please address the issues raised by reviewer 2 and submit another revision. I am sure this third revision will bring you close to acceptance for publication.

IMPORTANTLY, please ensure that an English expert has edited your revised manuscript for content and clarity. Reviewer 2 has kindly provided a marked-up version of your revision, but you should still seek out an English expert. Perhaps also seek out an expert in comparative genomics that may also improve the writing.

I look forward to seeing your revision, and thanks again for submitting your work to PeerJ.

Good luck with your revision,

-joe

Reviewer 1 ·

Basic reporting

In my previous review, I already noted that this manuscript was written in unacceptable English. I have just begun to read the present version of the text, and I again see that the language errors remain uncorrected. I believe this prevents me from further discussion of the manuscript. I am sure that the authors are unable to prepare an acceptable version of the text. I also do not think that constant submission of poorly prepared versions of the manuscript is the best strategy to get it published in any journal including PeerJ.

Experimental design

No comment

Validity of the findings

No comment

Additional comments

No comment

Reviewer 2 ·

Basic reporting

I am happy to see again the revised version of the manuscript about the mitogenome features and taxonomic position of A. xanthosoma. I feel that the text is mostly improved but I have highlighted some English mistakes and some modifications on the manuscript attached. Also, the section on the mitochondrial genome characterization and comparison of the manuscript still contains unnecessary repetitions (for example, presentation of tRNAs, rRNAs). It probably leads to need for the improvement of the flow and clarity of the relevant text. In my opinion, the manuscript should be reconsidered by an expert in this area, particularly in the field of insect mitochondrial genomics. Furthermore, I could not see the accession number of this newly sequenced mitogenome throughout the manuscript and it should be provided. There are also question regarding the presentation of the phylogenetic tree. As I stated before, Symphyta is a paraphyletic group and Orussoidea is thought to be a sister group with Apocrita. So I think it’s better to rotate the clade including Apocrita + Orussoidea + Cephoidea + Pamphilioidea in your phylogenetic tree to see the relationship between Orussoidea and Apocrita. The family and superfamily names could not been read in this new version of figure (Fig. 9) and should be improved.

Experimental design

No comment.

Validity of the findings

No comment

Additional comments

No comment

Annotated reviews are not available for download in order to protect the identity of reviewers who chose to remain anonymous.

---

## Round 0.4 · accepted · Accept

Dear Dr. Niu and colleagues:

Thanks for revising your manuscript based on the concerns raised by the reviewer. I now believe that your manuscript is suitable for publication, as this reviewer (as well as myself) feel you have addressed the major problems. Congratulations! There are few minor items to address (both linguistic and scientific), per the reviewer’s marked-up manuscript. Please handle these before sending your manuscript to production.

I look forward to seeing this work in print, and I anticipate it being an important resource for researchers studying hymenopteran genome structure and evolution. Thanks again for choosing PeerJ to publish such important work.

Best,

-joe

# Reviewer 2 ·

Basic reporting

I am happy to see again the revised version of the manuscript about the mitogenome features and taxonomic position of A. xanthosoma. I feel that the text is mostly improved. I have highlighted some minor changes on the manuscript attached.

Experimental design

No comment.

Validity of the findings

No comment.

Additional comments

No comment.

Annotated reviews are not available for download in order to protect the identity of reviewers who chose to remain anonymous.